# Women's decision-making autonomy and utilisation of maternal healthcare services: results from the Bangladesh Demographic and Health Survey

Bishwajit Ghose,[1] Da Feng,[1] Shangfeng Tang,[1] Sanni Yaya,[2] Zhifei He,[1] Ogochukwu Udenigwe,[2] Sharmistha Ghosh,[3] Zhanchun Feng[1]

BG and DF contributed equally.

[1]School of Medicine and Health Management, Tongji Medical College, Huazhong University of Science and Technology, Wuhan, China
[2]School of International Development and Global Studies, University of Ottawa, Ottawa, Canada
[3]Department of Sociology, University of Dhaka, Dhaka, Bangladesh

**Correspondence to**
Dr Shangfeng Tang;
sftang@hust.edu.cn

## ABSTRACT

**Objectives** The aim of this study was to determine the association between women's decision-making power and utilisation of maternal healthcare services (MHS) among Bangladeshi women.

**Settings** This is a nationally representative survey that encompassed Dhaka, Rajshahi, Rangpur, Chittagong, Khulna, Barisal and Sylhet in Bangladesh. Sample households were selected by a two-stage stratification technique. First, 207 clusters in urban areas and 393 in rural areas were selected for 600 enumeration areas with proportional probability. In the second stage, on average 30 households were selected systematically from the enumeration areas. Finally, 17 989 households were selected for the survey of which 96% were interviewed successfully.

**Participants** Cross-sectional data on 4309 non-pregnant women were collected from Bangladesh demographic and health survey 2014. Decision-making status on respondent's own healthcare, large household purchases, having a say on child's healthcare and visiting to family or relatives were included in the analysis.

**Results** Prevalence of at least four antenatal attendance, facility delivery and postnatal check-up were respectively 32.6% (95% CI 31.2 to 34), 40.6% (95% CI 39.13 to 42.07) and 66.3% (95% CI 64.89 to 67.71). Compared with women who could make decisions alone, women in the urban areas who had to decide on their healthcare with husband/partner had 20% (95% CI 0.794 to 1.799) higher odds of attending at least four antenatal visits and those in rural areas had 35% (95% CI 0.464 to 0.897) lower odds of attending at least four antenatal visits. Women in urban and rural areas had respectively 43% (95% CI 0.941 to 2.169) and 28% (95% CI 0.928 to 1.751) higher odds of receiving postnatal check-up when their health decisions were made jointly with their husband/partner.

**Conclusion** Neither making decisions alone, nor deciding jointly with husband/partner was always positively associated with the utilisation of all three types of MHS. This study concludes that better spousal cooperation on household and health issues could lead to higher utilisation of MHS services.

### Strengths and limitations of this study

► This is one of the few studies focusing on the correlation between women's decision-making autonomy and maternal healthcare utilisation in a South Asian country.
► Based on data from Bangladesh Demographic and Health Survey 2014, this study provides the most recent scenario of the utilisation of three key important components of maternal healthcare.
► Regional differences were observed in the prevalence of decision-making autonomy and utilisation of maternal healthcare services. However, the association was not a strong indication regarding the importance of decision-making autonomy for the uptake of maternal healthcare.
► The survey was cross-sectional. Therefore, it cannot affirm any causal inference or direction of the association.

## INTRODUCTION

There is a widespread consensus regarding the pivotal role of the utilisation of maternal healthcare services (MHS) in reducing maternal and child mortality and promoting women's reproductive health. Maternal mortality refers to deaths caused by pregnancy or childbirth-related complications. Since 2015, global maternal mortality rate (MMR) dropped by 44% at an average annual decline of 2.3%; however, it still remains the leading cause of death among adult women aged between 15 and 49 years.[1] The burden of maternal mortality is also disproportionately skewed towards the middle-income countries.[2] The most important causes of maternal mortality in middle-income countries are unsafe abortion, haemorrhage, eclampsia and obstructed labour as they together account for nearly two-thirds of total maternal mortality globally.[3 4] Growing consensus suggests that a vast majority of

these deaths are actually preventable simply by adopting the necessary precautions provisioned through basic MHS.[3 5]

The burden of maternal mortality is historically high in Bangladesh. However, the country has achieved noteworthy progress in terms of reducing MMR by three-quarters by 2015, as a part of its meeting the Millennium Development Goal 5A (MDG).[6] According to a study based on Bangladesh Maternal Mortality Surveys, maternal mortality was the largest single cause of death (20%) for women aged 15–49 years followed by malignancy and infectious diseases, and ranked third a decade later (14% deaths).[7] Despite the continued progress, the country is lagging far behind in ensuring universal access to reproductive health (MDG 5B), and the rate of utilisation of the basic MHS at the national level remains quite low.[8] According to Bangladesh Demographic and Health Survey (BDHS 2007), only about half of all mothers attended one or more antenatal visit and about one-fifth received at least one postnatal service. Mothers who do not attend antenatal care (ANC) services are also more unlikely to deliver at health facilities and receive postnatal services,[9] which increases the risk of pregnancy and childbirth-related complications.[5 9 10] The rate of health facility delivery is also notably low in Bangladesh with three-quarters of all births occurring at home and merely one-fifth are attended by a skilled birth attendant (SBA),[10] which is far below the internationally agreed target (90% births to be attended by SBA by 2015).[11]

Previous studies have attempted to explore the barriers to the utilisation of MHS, some from demographic, economic[9 12 13] and some from sociocultural and behavioural perspectives.[9 10 14 15] Apart from the socioeconomic aspects, there is also a growing number of study emphasising the role of women's decision-making autonomy on maternal health service utilisation and pregnancy outcomes.[16 17] However, the results remain somewhat mixed as some researches stress on the importance of wife's autonomy on making decisions and some proposing that joint decision-making by husbands/partners and wives can produce better reproductive health outcomes than when one partner is left behind from decision-making tasks. In the perspective of Bangladesh however, involvement of husbands/partners in decision making is particularly important because most families are male-headed and it is also the male figures who usually play the dominant role in important household decision making such as income expenditure and healthcare-related movement.[17] In South Asian countries including Bangladesh, gender discrimination and inequality remains a widespread phenomenon across various walks of life such as decision-making autonomy, intrahousehold resource allocation, property rights and access to healthcare.[18 19] Women's autonomy is a multidimensional concept which is hard to represent in a single definition. In short, it conveys a set of discrete components or phenomena essential for ensuring that women can exercise their rights with full potential. Therefore,

the aim of this study was to determine the association between women's decision-making power and utilisation of MHS among Bangladeshi women. For, this study, women's decision-making autonomy was measured across four different themes ranging from having a say in their own and children's healthcare decisions to household purchases and visiting family and relatives. Data were sourced from the latest BDHS survey which provides a large-scale quality data and representative of the general population.

## METHODS
### The survey: BDHS 2014
This is a cross-sectional study based on data from the Bangladesh demographic and health survey conducted in 2014. The 2014 survey was the sixth to be conducted in the country. This is a nationally representative survey that included both urban and rural areas encompassing all seven administrative divisions—Dhaka, Rajshahi, Rangpur, Chittagong, Khulna, Barisal and Sylhet. A division is a collection of districts (zilas), and each district is further divided into administrative units (upazilas), which are further divided into urban and rural areas. Sample households were selected by a two-stage stratification technique. First, 207 clusters in urban areas and 393 in rural areas were selected for 600 enumeration areas with proportional probability. In the second stage, on average 30 households were selected systematically from the enumeration areas. Finally, 17989 households were selected for the survey of which 96% were interviewed successfully. Details on the survey and sampling technique are available in the final report.

### Variables
Outcome variable: the outcome variables chosen for this study were three basic types of MHS offered by the healthcare system in Bangladesh: 1) ANC services, 2) facility delivery services and 3) postdelivery check-up services. Information on these topics were collected by face-to-face interview with the respondents. Women were asked the number of times they received ANC, and the frequency ranged from '0' to '20'. However, for this study, ANC was categorised as adequate (4/4+) and inadequate (<4) as per the WHO recommendation, which suggests at least four ANC attendance during pregnancy. Place of delivery was categorised as 'facility delivery' and 'delivery at home'. Facility delivery included delivery in public or private hospitals or clinics, NGO clinics. The third outcome variable, postdelivery check-up services, was categorised as yes (for those who received any postnatal check-up) and no (for those who did not receive any postnatal check-up).

Explanatory variables of interest were women's decision-making power on the following four themes: 1) person who usually decides on respondent's healthcare, 2) person who usually decides on large household purchases, 3) final say on: child's healthcare, 4) person

who usually decides on visits to family or relatives. In types of decision-making tasks, a joint decision by women and their husband was highest. Possible answers were respondent alone, respondent and husband/partner jointly, husband/partner alone and other. The categories were collapsed into three by combining the last two into one (husband/partner alone and other).

The covariates included in the analysis were age: 15–20/21–24/25–29/30+ years; Educational attainment: no education/incomplete primary/complete primary/ incomplete secondary/complete secondary/higher; currently working: no/yes; wealth index: poorer/middle/ richer/richest/poorest; parity: 1/2/3/3+.

## DATA ANALYSIS

Datasets were checked for missing values and outliers and weighted prior to analysis. Basic sociodemographic variables were described by descriptive statistics. Chi-square bivariate tests were performed to examine the group differences (utilisation vs non-utilisation of MCHs) for all the explanatory variables. The variables that showed significance at $p \leq 0.25$ in the bivariate tests were retained for final regression analysis. The association between utilisation of MCHs and the independent variables was measured by binary logistic regression. Three separate regression models were run for each of the outcome variables. Results of the regression analysis were presented as adjusted ORs (AOR) with corresponding 95% CIs. The outcomes of the regression analysis were reported in terms of AOR and corresponding 95% CIs. Model fitness was verified by the Hosmer-Lemeshow goodness-of-fit test. All tests were two-tailed and was considered significant at 5%. Data were analysed using SPSSV.22.

## ETHICS

All participants gave informed consent prior to taking part in the voluntary interview. The survey was approved by the ICF International Institutional Review Board, who is responsible for reviewing the procedures and questionnaires for standard DHS surveys.

## RESULTS
### Population characteristics

Table 1 shows that majority of participants belonged to the youngest age groups of 15–20 years. About one-third of the women were from urban (32.5%) areas which is similar to the country's level scenario; 13.3% of the women had no formal education and 11.4% had completed primary level of education. Rate of illiteracy was high among rural women compared with their urban counterparts (9.8% vs 14.9%). Rate of completion of secondary was 7.5% and 11.8% had higher than secondary level education. Only about one-fifth of the women reported having an employment, and urban women had slightly higher rate of employment (22.9% vs 19.0%) than rural

**Table 1** Basic characteristics of the study population (n=4309), Bangladesh Demographic and Health Survey 2014

| Variables | N (%) | Urban 1381 (32.5) | Rural 2873 (67.5) |
|---|---|---|---|
| **Age (years)** | | | |
| 15–20 | 1178 (27.7) | 25.3 | 28.8 |
| 21–24 | 1144 (26.9) | 28.7 | 26.0 |
| 25–29 | 1091 (25.6) | 27.1 | 25.0 |
| 30+ | 841 (19.8) | 18.9 | 20.2 |
| **Educational attainment** | | | |
| No education | 564 (13.3) | 9.8 | 14.9 |
| Incomplete primary | 658 (15.5) | 12.8 | 16.7 |
| Complete primary | 487 (11.4) | 10.2 | 12.0 |
| Incomplete secondary | 1724 (40.5) | 38.6 | 41.5 |
| Complete secondary | 319 (7.5) | 8.9 | 6.8 |
| Higher | 502 (11.8) | 19.6 | 8.0 |
| **Currently working** | | | |
| No | 3333 (78.3) | 81.0 | 77.1 |
| Yes | 921 (21.7) | 19.0 | 22.9 |
| **Wealth index** | | | |
| Poorer | 806 (18.9) | 8.6 | 26.2 |
| Middle | 814 (19.1) | 7.8 | 24.3 |
| Richer | 901 (21.2) | 12.0 | 22.6 |
| Richest | 860 (20.2) | 28.4 | 17.7 |
| Poorest | 873 (20.5) | 43.2 | 9.2 |
| **Parity** | | | |
| 1 | 1700 (40.0) | 44.6 | 37.7 |
| 2 | 1286 (30.2) | 32.2 | 29.3 |
| 3 | 664 (15.6) | 13.5 | 16.6 |
| 3+ | 604 (14.2) | 9.7 | 16.4 |

women. Majority of the women belonged to the highest wealth quintile (20.5%) and a little less than one-fifth in the poorest wealth quintile (18.9%). A wide wealth disparity was observed between participants in urban and rural areas as 43.2% of the women in the highest wealth quintile were from urban areas compared with only 9.2% from rural areas. Two-fifth of the women had only one child and 14.2% had more than three children.

Based on the availability of on the dataset, four types of decision-making tasks were considered relevant to MCH in this study: 1) person who usually decides on respondent's healthcare, 2) person who usually decides on large household purchases, 3) final say on: child's healthcare, 4) person who usually decides on visits to family or relatives. For all types of decision-making tasks, a joint decision by women and their husband was highest. Table 2 shows that frequency of having autonomy in all types of the decisions was lower among rural women except for final say on child's healthcare. In majority of the cases,

**Table 2** Women's household decision-making characteristics, Bangladesh Demographic and Health Survey 2014

| Types of decision making | Respondent alone | | Respondent and husband/partner | | Husband/partner alone/other | |
|---|---|---|---|---|---|---|
| | Urban | Rural | Urban | Rural | Urban | Rural |
| Person who usually decides on respondent's healthcare | 12.2 | 12 | 51.3 | 47.2 | 36.5 | 40.9 |
| Person who usually decides on large household purchases | 7.1 | 5.4 | 53.4 | 46.5 | 39.5 | 48.0 |
| Final say on: child's healthcare | 14.2 | 15.2 | 58.8 | 54.2 | 27.0 | 30.6 |
| Person who usually decides on visits to family or relatives | 8.3 | 7.4 | 53.4 | 47.3 | 38.3 | 45.2 |

decisions were made jointly by women and the husband/partner. Husbands/partners had notably higher rate of autonomy than women in making these decisions in both rural and urban areas.

Table 3 shows the prevalence of availing the three types of MHS stratified by place of residency. Prevalence of ANC attendance, facility delivery and postnatal check-up were respectively 32.6%, 40.6% and 66.3% (not shown in the table). Results of cross-tabulation show that the rate of utilisation of these services were higher among urban women compared with their rural counterparts, higher among women aged between 21 and 24 years, having incomplete secondary level schooling, living in the richest households, currently not working and had given birth only once. In majority of the cases, women who could make the decisions jointly with husband/partner were more like to enjoy the MCH services.

### Association between decision-making ability and utilisation of MCH

Results of regression analysis on the association between decision-making ability and utilisation of MCH are presented in table 4.

In the urban areas, women who could decide their healthcare with husband/partner had 20% (95% CI 0.794 to 1.799) higher odds of attending at least four ANC compared with those who could make decisions alone. In the rural areas however, women who could make decisions alone were 35% (95% CI 0.464 to 0.897) less likely to do so. The odds of delivering at a health facility were 25% (95% CI 0.888 to 1.748) higher among rural women who made own health decisions jointly with husband/partner. Women in urban and rural areas had respectively 43% (95% CI 0.941 to 2.169) and 28% (95% CI 0.928 to 1.751) higher odds of receiving postnatal check-up when they made their health decisions jointly with husband/partner. Women in urban and rural areas who had less autonomy on deciding large household purchases were respectively 28% (95% CI 0.384 to 1.365) and 20% (95% CI 0.492 to 1.285) less likely to have at least four ANC visits. Rural women who had to decide on large household purchases with husband/partner had 15% (95% CI 0.547 to 1.332) lower odds of receiving postnatal check-up.

Having autonomy in deciding children's healthcare did not show noticeable impact on receiving ANC services. Odds of receiving postnatal check-up were respectively 22% (95% CI 0.503 to 1.212) and 31% (95% CI 0.501 to 0.946) lower and facility delivery respectively 11% (95% CI 0.574 to 1.413) and 12% (95% CI 0.577 to 1.266) lower among urban and rural women who had to make the decisions jointly with husband/partner. In urban areas, women who did not have the autonomy to decide on visiting family or relatives alone were 18% (95% CI 0.491 to 1.362) less likely to attend at least four antenatal visits. The odds of receiving postnatal check-up were respectively 32% (95% CI 0.760 to 2.311) and 11% (95% CI 0.757 to 1.636) higher among urban and rural women who could decide on visiting family or relatives jointly with husband/partner.

### DISCUSSION AND CONCLUSION
### Main findings

Based on a nationally representative data from BDHS, this study explored the association between women's decision-making power and utilisation of ANC, facility delivery and postnatal health check-up among adult non-pregnant women aged between 15 and 49 years in Bangladesh. Our results show that the prevalence of ANC attendance, facility delivery and postnatal check-up were respectively 32.6%, 40.6% and 66.3%, which indicates a considerable improvement compared with the previous estimates. In urban and rural areas respectively, the rate of attending at least four antenatal visits increased from 36.7% and 11.7% in 2004 (44.8% and 19.8% in 2011) to 46.1% and 26% in 201[20]. Utilisation of health facility delivery increased from 12% in 2004 (>29% in 2011) to >40% in 2014,[21] and postnatal check-up of mothers increased from 27.3% to >66% during the same period.[22]

Compared with women who decided on their healthcare alone, those who decided jointly with husband/partner had higher likelihood of using all three types of services (except for antenatal visits among rural women). However, women could decide large household purchases alone had higher likelihood of attending at least four antenatal visits. Similar association was observed for utilisation of postnatal care among women in rural but not urban areas. Having decision-making autonomy on child's healthcare showed significant association with the utilisation of facility delivery and postnatal check-ups but not antenatal visits. Having decision-making autonomy on visiting family/relatives showed significant association with the utilisation of postnatal check-ups but not antenatal visits and facility delivery.

**Table 3** Percentage of women who reported using three types of MCH across the explanatory variables, Bangladesh Demographic and Health Survey 2014

| | ANC | | Facility delivery | | Health check-up after birth | |
|---|---|---|---|---|---|---|
| | **Urban (46.1)** | **Rural (26)** | **Urban (42.1)** | **Rural (67.7)** | **Urban (79.4)** | **Rural (60.0)** |
| Age (years) | | | | | | |
| 15–20 | 29.0 | 26.7 | 28.2 | 27.1 | 29.4 | 26.0 |
| 21–24 | 27.9 | 27.4 | 29.2 | 25.2 | 27.6 | 24.4 |
| 25–29 | 19.6 | 15.8 | 19.6 | 17.5 | 18.7 | 19.5 |
| 30+ | 23.4 | 30.1 | 23.0 | 30.1 | 24.4 | 30.0 |
| p Value | 0.478 | **0.005** | **0.051** | **0.101** | 0.322 | 0.291 |
| Educational attainment | | | | | | |
| No education | 5.2 | 6.8 | 5.0 | 6.8 | 8.6 | 10.9 |
| Incomplete primary | 7.7 | 12.2 | 9.3 | 10.1 | 11.5 | 14.7 |
| Complete primary | 7.4 | 8.6 | 6.3 | 9.7 | 8.8 | 11.3 |
| Incomplete secondary | 36.1 | 47.9 | 39.8 | 45.3 | 38.6 | 43.2 |
| Complete secondary | 11.0 | 10.7 | 11.1 | 10.9 | 9.9 | 8.6 |
| Higher | 32.7 | 13.9 | 28.5 | 17.2 | 22.7 | 11.4 |
| p Value | **<0.001** | **<0.001** | **<0.001** | **<0.001** | **<0.001** | **<0.001** |
| Wealth index | | | | | | |
| Poorest | 5.2 | 15.6 | 3.1 | 12.6 | 5.7 | 20.0 |
| Poorer | 4.4 | 19.7 | 5.8 | 17.4 | 6.3 | 21.9 |
| Middle | 7.4 | 21.9 | 8.1 | 25.3 | 10.5 | 24.2 |
| Richer | 25.0 | 25.9 | 25.8 | 25.9 | 28.0 | 21.2 |
| Richest | 58.1 | 16.8 | 57.2 | 18.7 | 49.5 | 12.8 |
| p Value | **<0.001** | **<0.001** | **<0.001** | **<0.001** | **<0.001** | **<0.001** |
| Currently working | | | | | | |
| No | 82.1 | 79.0 | 83.6 | 81.1 | 81.7 | 74.8 |
| Yes | 17.9 | 21.0 | 16.4 | 18.9 | 18.3 | 25.2 |
| p Value | 0.336 | **0.157** | **0.004** | **<0.001** | 0.204 | **<0.001** |
| Parity | | | | | | |
| 1 | 50.1 | 44.0 | 50.2 | 49.1 | 46.3 | 42.2 |
| 2 | 34.7 | 32.1 | 33.0 | 29.2 | 33.3 | 28.7 |
| 3 | 10.0 | 14.7 | 11.0 | 13.8 | 12.4 | 15.5 |
| 4 | 5.2 | 9.2 | 5.8 | 8.0 | 8.0 | 13.7 |
| p Value | **<0.001** | **<0.001** | **<0.001** | **<0.001** | **<0.001** | **<0.001** |
| Decides on own healthcare | | | | | | |
| Alone | 11.8 | 15.0 | 13.1 | 11.8 | 12.2 | 12.0 |
| Jointly | 53.7 | 49.3 | 51.8 | 49.6 | 52.6 | 48.1 |
| Husband/other | 34.5 | 35.7 | 35.0 | 38.5 | 35.1 | 39.9 |
| p Value | 0.246 | **<0.001** | 0.291 | **0.165** | **0.096** | 0.418 |
| Decides on large household purchases | | | | | | |
| Alone | 7.8 | 6.8 | 8.1 | 5.5 | 7.7 | 5.8 |
| Jointly | 55.6 | 46.9 | 52.3 | 47.6 | 53.8 | 47.5 |
| Husband/other | 36.6 | 46.3 | 39.5 | 46.9 | 38.5 | 46.7 |
| p Value | **0.108** | **0.117** | **0.189** | **0.144** | **0.149** | **0.006** |
| Decides on child's healthcare | | | | | | |

**Table 3**   Continued

| | ANC | | Facility delivery | | Health check-up after birth | |
|---|---|---|---|---|---|---|
| | **Urban (46.1)** | **Rural (26)** | **Urban (42.1)** | **Rural (67.7)** | **Urban (79.4)** | **Rural (60.0)** |
| Alone | 13.8 | 17.8 | 15.0 | 16.3 | 14.6 | 16.2 |
| Jointly | 61.9 | 55.1 | 59.4 | 54.9 | 59.4 | 54.9 |
| Husband/other | 24.3 | 27.1 | 25.5 | 28.8 | 26.0 | 28.9 |
| p Value | **0.077** | **0.015** | **0.175** | **0.219** | **0.231** | **0.035** |
| Decides on visits to family or relatives | | | | | | |
| **Alone** | 8.6 | 8.0 | 9.9 | 7.2 | 8.3 | 7.7 |
| **Jointly** | 57.5 | 46.7 | 53.6 | 47.9 | 54.3 | 47.2 |
| **Husband/other** | 33.9 | 45.3 | 36.5 | 44.9 | 37.4 | 45.1 |
| p Value | **0.007** | **0.140** | **0.029** | **0.102** | **0.075** | **0.031** |

### Comparison with existing literature

Results indicate that in majority of the cases decisions were made jointly followed by men alone and women alone. A previous study conducted on South Asian countries reported a similar situation that women's healthcare decision were made without their participation in Nepal

**Table 4**   Association between decision-making ability and utilisation of MCH in Bangladesh, Bangladesh Demographic and Health Survey 2014

| | Antenatal care OR (95% CI) | | Delivery at a health facility OR (95% CI) | | Health check-up after delivery OR (95% CI) | |
|---|---|---|---|---|---|---|
| | **Urban** | **Rural** | **Urban** | **Rural** | **Urban** | **Rural** |
| Decides on own healthcare | | | | | | |
| Alone | 1 | 1 | 1 | 1 | 1 | 1 |
| Jointly | 1.195 (0.794 to 1.799) | 0.645 (0.464 to 0.897) | 0.996 (0.606 to 1.327) | 1.246 (0.888 to 1.748) | 1.428 (0.941 to 2.169) | 1.275 (0.928 to 1.751) |
| Husband/other | 1.087 (0.806 to 1.750) | 0.983 (0.635 to 1.227) | 1.072 (0.731 to 1.572) | 1.001 (0.703–1.424) | 1.052 (0.662 to 1.671) | 1.035 (0.815–1.579) |
| Decides on large household purchases | | | | | | |
| Alone | 1 | 1 | 1 | 1 | 1 | 1 |
| Jointly | 0.724 (0.384 to 1.365) | 0.795 (0.492–1.285) | 1.050 (0.621 to 1.776) | 0.997 (0.629–1.581) | 1.02 (0.251 to 1.745) | 0.854 (0.547–1.332) |
| Husband/other | 0.970 (0.361 to 1.444) | 0.805 (0.493–1.315) | 0.734 (0.420 to 1.282) | 0.924 (0.587–1.455) | 0.950 (0.617 to 1.935) | 0.943 (0.548–1.497) |
| Decides on child's healthcare | | | | | | |
| Alone | 1 | 1 | 1 | 1 | 1 | 1 |
| Jointly | 0.978 (0.641 to 1.491) | 0.983 (0.558–1.499) | 0.897 (0.574 to 1.413) | 0.884 (0.577–1.266) | 0.781 (0.503 to 1.212) | 0.688 (0.501–0.946) |
| Husband/other | 1.100 (0.751 to 1.612) | 0.930 (0.696–1.243) | 1.079 (0.713 to 1.635) | 0.945 (0.620–1.452) | 0.898 (0.562 to 1.384) | 0.949 (0.645–1.617) |
| Decides on visits to family or relatives | | | | | | |
| Alone | 1 | 1 | 1 | 1 | 1 | 1 |
| Jointly | 0.818 (0.491 to 1.362) | 1.018 (0.645–1.727) | 0.981 (0.664 to 0.178) | 1.108 (0.776–1.729) | 1.325 (0.760 to 2.311) | 1.113 (0.757–1.636) |
| Husband/other | 1.063 (0.701 to 1.928) | 1.078 (0.709–1.640) | 0.887 (0.421 to 1.121) | 1.075 (0.711–1.625) | 1.050 (0.819 to 1.567) | 0.943 (0.645–1.378) |

Adjusted for the variables found significant in the bivariate test in table 3.

(72.7%), Bangladesh (54.3%) and India (48.5%).[23] Regarding the association between decision-making autonomy and MHS utilisation, comparison between the findings of the present study with the existing ones requires consideration of several important issues. First, different studies uses different indicators of women's decision-making autonomy and different types of MHS. Moreover, some studies report involvement of various family members and not just women and husbands/partners. Regardless of that, our findings have consistent and conflicting points with previous ones. Low level of women's autonomy was found to be a contributing factor to poor maternal health service utilisation in Nepal,[24] India,[25] but not in Kenya.[26] In Ethiopia, decision-making autonomy on place of birth showed a positive association with utilisation of institutional delivery.[27]

While women's lack of decision-making autonomy can be attributed to poor utilisation of MHS, it however should not be ignored that autonomy in certain circumstances can also result in less spousal communication and low male involvement in reproductive care. Growing number of studies indicate that inadequate spousal communication and male involvement in reproductive care are associated with poor reproductive and sexual health consequences, and recommend policies to promote spousal communication and cooperation for improved maternal health outcomes.[28 29] In Nepal for instance, economic autonomy among women was associated with lower likelihood of couple communication during pregnancy, while domestic decision-making autonomy was associated with both lower likelihood of intraspousal communication during pregnancy and husband's presence at antenatal visits.[30] Husbands' involvement in ANC has been shown to have a positive influence on utilisation of antenatal visits in Ethiopia.[27] Husbands' involvement was also associated with utilisation of professional care during delivery in rural Bangladesh and India.[31]

In light of the above-mentioned discussion, it is suggestible that health projects aiming to improve the utilisation of MHS should try to focus on women's autonomy and at the same time promote male involvement in women's reproductive care. A qualitative study on male participation in reproductive health in Bangladesh reported poor interaction between husband and wife regarding sexual reproductive health issues which makes it difficult for men to recognise the reproductive health issues of women.[32] The study also reported that men do not feel comfortable to take their wives to the health facility, which suggests the presence of complex social and cultural factors preventing effective spousal communication regarding reproductive health issues. In the traditionally male-dominated society in Bangladesh where male figures are usually involved in family decision making, excluding men from maternal health decision-making issues could prevent men from making informed decision for their wives/partners.

This study has several limitations. First, this study included only four aspects of women's decision making. Thus, the findings do not indicate women's overall mobility and empowerment but rather specifically focuses on a limited range of indicators. As the participants were only women, there remains a potential for bias/discordance regarding the level of autonomy enjoyed by women as this is to a large extent a subjective phenomenon. Arguably, collecting information from both men and women could generate more a reliable picture on women's mobility and empowerment. So the association between women's autonomy and healthcare service use may be underestimated when only women's reports are considered.[30] In addition, spousal autonomy is a complex concept and difficult to quantify and there is no universally agreed definition or tool for measurement. Last but not least, utilisation status of MHS was reported by women and was not verified from medical records, and therefore subject to recall bias.

**Acknowledgements** The authors would like to express sincere thanks to DHS for distributing the datasets that made this study possible.

**Contributors** BG and DF contributed equally to this work. ST, BG and ZF designed the study. DF, BG and ST conducted the analysis and wrote the paper. DF, SY, HZF, SG, OU participated in editing of the paper. All authors discussed the results and approved the revision of the final manuscript.

**Funding** This research was supported by the National Natural Science Foundation of China (71473097 and 71273097).

**Competing interests** None declared.

**Ethics approval** The survey was approved by the ICF International Institutional Review Board (IRB) who is responsible for reviewing the procedures and questionnaires for standard DHS surveys.

**Provenance and peer review** Not commissioned; externally peer reviewed.

**Data sharing statement** Data are available from DHS website through registration.

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
