## [Reviewer comments · BMJ Open]

ARTICLE DETAILS

TITLE (PROVISIONAL)	Women's decision making autonomy and utilization of maternal healthcare services: results from the Bangladesh Demographic and Health Survey
AUTHORS	Ghose, Bishwajit; Feng, Da; Tang, Shangfeng; Yaya, Sanni; He, Zhifei; Udenigwe, Ogochukwu; Ghosh, Sharmistha; Feng, Zhanchun

VERSION 1 – REVIEW

REVIEWER	Xu Aijun Nanjing University of Chinese Medicine, Health Economics and Management School P. R. China
REVIEW RETURNED	25-Apr-2017

GENERAL COMMENTS	This paper tried to study the relationship between decision making autonomy and utilization of maternal healthcare services for women in Bangladesh. It's a good paper if some parts can be showed more clearly and completely. 1. Abstract shows 4309 not-pregnant women were chosen as participants. These information however could not be found in the body of paper.2. The information between the lines in Result-Population characteristics section does not match the table1. The results of urban and rural seems to be mistaken.3. In table 3, we have four groups for age variables but five groups for cross-tabulation results. The wording does not match the table result regarding Currently working.4. Few wording need to be checked .
--

REVIEWER	Eric Arthur Kwame Nkrumah University of Science and Technology, Kumasi, Ghana.
REVIEW RETURNED	22-May-2017

GENERAL COMMENTS	1. The authors must revise the paper and correct the grammar.2. The authors must perform the regression again. Even though the authors use a justification that variables that are not significant at the univariate stage are not included in the regression, this is not appropriate for such studies. Variables may not be significant individually in influencing a particular outcome but may be significant jointly in which case it is not appropriate not to include such
--

	variables in the regression. In short, I suggest that the authors include relevant variables such as wealth, education, age, occupation and other variables that can influence the demand for MCH services in the regression. So many studies have illustrated the importance of such variables.
--	--

VERSION 1 – AUTHOR RESPONSE

Reviewer: 1

Reviewer Name: Xu Aijun

Institution and Country: Nanjing University of Chinese Medicine, Health Economics and Management School, P. R. China

Please state any competing interests or state 'None declared': None declared

Please leave your comments for the authors below

This paper tried to study the relationship between decision making autonomy and utilization of maternal healthcare services for women in Bangladesh. It's a good paper if some parts can be showed more clearly and completely.

1. Abstract shows 4309 not-pregnant women were chosen as participants. These information however could not be found in the body of paper.

- Sample size was now mentioned in table 1.

2. The information between the lines in Result-Population characteristics section does not match the table1. The results of urban and rural seems to be mistaken.

- These errors were corrected. Thank you!

3. In table 3, we have four groups for age variables but five groups for cross-tabulation results. The wording does not match the table result regarding Currently working.

- Table 3 was corrected as suggested. Thank you!

4. Few wording need to be checked .

- The paper was checked by a native English author for language issues.

Reviewer: 2

Reviewer Name: Eric Arthur

Institution and Country: Kwame Nkrumah University of Science and Technology, Kumasi, Ghana.

Please state any competing interests or state 'None declared': None Declared

Please leave your comments for the authors below

1. The authors must revise the paper and correct the grammar.

- The paper was checked by a native English author for language issues.

2. The authors must perform the regression again. Even though the authors use a justification that variables that are not significant at the univariate stage are not included in the regression, this is not appropriate for such studies. Variables may not be significant individually in influencing a particular outcome but may be significant jointly in which case it is not appropriate not to include such variables in the regression. In short, I suggest that the authors include relevant variables such as wealth, education, age, occupation and other variables that can influence the demand for MCH services in the regression. So many studies have illustrated the importance of such variables.

- Thanks very much for this comment. We do agree with you, and hence used the most generous threshold for significance in the chi-square tests ($p < 0.25$ instead of 0.05) to ensure no potentially important variable is ignored. By this criteria, all the variables qualified for inclusion in all three

regression models, except for age in the third one (post natal check). We reanalysed the model by including age, however it did not make any noticeable alteration in the association between the outcome and explanatory variables.

VERSION 2 – REVIEW

REVIEWER	Eric Arthur Kwam Nkrumah University of Science and Technology Kumasi, Ghana.
REVIEW RETURNED	28-Jun-2017

GENERAL COMMENTS	The authors should inform readers about the limitations of the study.
---

VERSION 2 – AUTHOR RESPONSE

Thanks for the valuable comments. First, we revised the title slightly to: 'Women's decision-making autonomy and utilization of maternal healthcare services: results from the Bangladesh Demographic and Health Survey' Second, we stated any competing interests or state 'None declared' Third, we informed readers about the limitations of the study in the last paragraph.